# Investigation of the Long-Term Stability of Different Polymers and Their Blends with PEO to Produce Gel Polymer Electrolytes for Non-Toxic Dye-Sensitized Solar Cells

Marius Dotter *, Jan Lukas Storck, Michelle Surjawidjaja, Sonia Adabra and Timo Grothe

Faculty of Engineering and Mathematics, Interaktion 1, Bielefeld University of Applied Sciences, 33619 Bielefeld, Germany; jan_lukas.storck@fh-bielefeld.de (J.L.S.); michelle.surjawidjaja@fh-bielefeld.de (M.S.); sonia.adabra@fh-bielefeld.de (S.A.); timo.grothe@fh-bielefeld.de (T.G.)
* Correspondence: marius.dotter@fh-bielefeld.de; Tel.: +49-521-106-70934

**Abstract:** The electrolyte for dye-sensitized solar cells (DSSCs) is subject of constant innovation, as the problems of leakage and drying greatly reduce the long-term stability of a device. One possible way to solve these problems is the use of gel polymer electrolytes (GPEs) with a gelling structure, which offer different advantages based on the used polymers. Here, potential GPE systems based on dimethyl sulfoxide (DMSO) as solvent for low-cost, non-toxic and environmentally friendly DSSCs were investigated comparatively. In order to observe a potential improvement in long-term stability, the efficiencies of DSSCs with different GPEs, consisting of polyacrylonitrile (PAN), acrylonitrile-butadiene-styrene (ABS), polyvinyl alcohol (PVA) and poly (vinylidene fluoride) (PVDF) and their blends with poly (ethylene oxide) (PEO), were investigated over a period of 120 days. The results indicate that blending the polymers with PEO achieves better results concerning long-term stability and overall efficiency. Especially the mixtures with PAN and PVDF show only slight signs of deterioration after 120 days of measurement.

**Keywords:** dye-sensitized solar cells (DSSC); gel polymer electrolyte; long-term stability; dimethyl sulfoxide (DMSO); poly (ethylene oxide) (PEO); natural dyes; non-toxic; polymers



## 1. Introduction

In the context of climate change and severe problems regarding fossil fuel and nuclear waste, harvesting of solar energy is a potential alternative to fulfil future energy needs as it is a rapidly growing renewable energy technology [1]. One challenge concerning conventional semiconductor-based photovoltaics are their relatively high production costs due to the need of a cleanroom and uncommon elements like indium [2,3]. Therefore dye-sensitized solar cells (DSSCs), fabricated in a simple process without a cleanroom and from inexpensive materials, offer a good alternative [1,4,5]. The environmental friendliness of DSSCs can be further improved by using non-toxic materials, such as widely available natural dyes [4,5].

In addition to improving the conversion efficiency, which is significantly lower when using natural dyes as absorbers compared to conventional toxic ones, it is also generally crucial to improve the long-term stability, which is commonly deteriorated by leakage of liquid electrolyte [4–6]. Generally, the electrolyte closes the electric cycle of DSSCs via iodine-triiodide as redox shuttle transferring an incoming electron from the counter electrode and regenerating the oxidized dye with it [6,7]. The leakage problem is particularly pronounced in textile-based DSSCs, which are promising in terms of textile architecture and wearable electronics, due to the open-pore structure of the textile and the inherent sealing inability [6,8].

In order to hinder solvent evaporation or leakage of the electrolyte and to enhance the long-term stability, gelling the electrolyte with polymers is feasible [7,9,10]. The solvent of

a gel polymer electrolyte (GPE) obtained in this way has classically a concentration above 50 wt% and is trapped by the added polymer, which forms a three-dimensional network wherein ion movement in the remaining liquid phase still occurs [11–13]. With regard to the used polymer, it is commonly distinguished between inert ones, where ions move only in the trapped salt solution, and coordinating polymer frameworks, in which the cation is dissolved by the polymer and thus the solid phase contributes also to the ion transport [12]. Examples regarding polymers for inert GPEs are poly (methyl methacrylate) (PMMA), polyacrylonitrile (PAN) or poly (vinylidene fluoride) (PVDF) [9,12]. Predominantly, GPEs for DSSCs apply poly (ethylene oxide) (PEO), which is a coordinating polymer [7,9]. The crystallinity of PEO is known to hinder ionic movement, wherefore blending with other polymers can be conducted to reduce the crystallinity by imposing some disorder within the structure [12,14].

Due to our motivation to produce non-toxic DSSCs, most frequently used toxic solvents like acetonitrile are out of the question [7]. Therefore, an important criterion for the polymers used in this paper is their solvability in the low-toxic solvent dimethyl sulfoxide (DMSO), which proved to be suitable regarding GPEs of environmentally friendly DSSCs [15].

PAN is very well soluble in DMSO [16] and often used as a basis for GPE DSSCs [17–19]. Like mentioned above, PAN provides an inert ion transport and offers ionic conductivity higher than PEO [19]. Furthermore, as mentioned by Chen et al. [17], PAN can be used for gelling liquid electrolytes and consequently should provide a suitable polymer matrix for a long-term stable GPE.

Another polymer used in an electrolyte blend with PMMA for batteries is acrylonitrile-butadiene-styrene (ABS), where it was inserted to improve the mechanical strength [20,21]. It should be mentioned that it is unfortunately not possible to dissolve PMMA in DMSO, so we do not use this polymer and instead mixed ABS with PEO as an alternative. Additionally, to the best of our knowledge, ABS has not previously been used as a polymer regarding GPEs of DSSCs.

As a further polymer, the synthetically produced polyvinyl alcohol (PVA) is used, which is considered non-toxic and well dissolvable in DMSO [22]. In addition to its good adhesive and film-forming properties, PVA is noted for its stability and excellent mechanical strength [22,23].

The last polymer tested was PVDF, with and without the addition of PEO as GPE. In appropriate dosage, PVDF enhances the viscosity and conductivity, leading to an enhanced cell efficiency [24,25]. It is also expected that the presence of PVDF reduces the recombination rate at the semiconductor/polymer interface and therefore lowers the internal resistance of the DSSC [25].

In this paper, GPE systems based on various polymers are comparatively evaluated with respect to their suitability for enhancing the long-term stability of environmentally friendly DSSCs with natural dyes. Therefore, current–voltage characteristics (I-U curves) of the GPE DSSCs and a reference with commercial liquid electrolyte were investigated over 120 days and the efficiencies were calculated to enable comparison.

A detailed characterization of the various GPEs was not done, as this would exceed the scope of the paper. As a second point, the purpose of this paper is to identify interesting GPEs, which can then be examined in more detail and optimized further. Furthermore, value was placed on a simple, consistent design of the remaining components of the DSSC to increase comparability. Due to the focus on non-toxic, low-cost materials, which is the reason for low efficiencies, a comparison with significantly more complex DSSCs is, of course, not possible.

## 2. Materials and Methods

Glass electrodes coated with fluorine-doped tin oxide, whereby the front electrodes were additionally pre-coated with a $TiO_2$ layer, and the liquid electrolyte for the reference sample, were purchased from Man Solar (Petten, The Netherlands). The exact formulation

of the reference electrolyte is unfortunately not known. To gain a catalyst layer on the counter electrodes, graphite was applied using a graphite pencil with a hardness of 6B (Faber-Castell, Stein, Germany), which is a common technique [26]. The semiconductor layers on the front electrodes were dyed by inserting them in a solution containing natural anthocyanins for 10 min. This solution was previously extracted from 10 g forest fruit tea (Mayfair, Wilken Tee GmbH, Fulda, Germany), which proved to be a suitable and low-cost source of the natural dyes [5], in 120 mL distilled water by stirring for 15 min at room temperature with a filtering step thereafter. After dyeing of the $TiO_2$ layer, the front electrodes were rinsed with distilled water and dried at room temperature.

To gain a polymer blend, 600,000 g/mol PEO from S3 Chemicals (Bad Oeynhausen, Germany) was mixed with PAN as L-PAN, kindly provided by Dralon GmbH (Dormagen, Germany), ABS (3D printing filament, Filamentworld, Neu-Ulm, Germany), PVA under the brand name GOHSENOL$^{TM}$ (NIPPON GOHSEI Europe GmbH, Düsseldorf, Germany), and PVDF with Amboflon® as the brand name (Ambofluor GmbH Co. KG, Hamburg, Germany). For comparison, these polymers were also utilized without PEO as the basis for GPEs. The composition of each sample is depicted in Table 1.

**Table 1.** Composition of investigated GPEs. For each sample, three DSSCs (labeled with "a", "b" and "c") were assembled with the corresponding electrolyte.

| Sample Number | Main Polymer (wt%) | PEO (wt%) | Electrolytic Salts (KI + $I_2$) (wt%) |
|---|---|---|---|
| 1 | PAN, 2.4 | 9.5 | 8.9 |
| 2 | PAN, 1.4 | 5.4 | 5.1 |
| 3 | PAN, 13.8 | 0 | 10.3 |
| 4 | ABS, 2.5 | 10.1 | 9.4 |
| 5 | ABS, 17.4 | 0 | 13.0 |
| 6 | PVA, 1.9 | 7.5 | 7.0 |
| 7 | PVA, 13.8 | 0 | 10.3 |
| 8 | PVDF, 3.2 | 9.1 | 8.8 |
| 9 | PVDF, 7.1 | 0 | 5.3 |
| Reference | - | - | Man Solar electrolyte |

Concerning the here-presented GPEs, the iodide-triiodide redox couple was based on potassium iodide (KI) purchased from VWR (Darmstadt, Germany) mixed at a weight ratio of 2:1 with molecular iodine ($I_2$) purchased from Carl Roth (Karlsruhe, Germany), inspired by Lugol's solution. These electrolytic salts were dissolved in DMSO (Carl Roth, Karlsruhe, Germany). For each sample depicted in Table 1, the related polymers were subsequently added. The components of the GPEs were stirred at 70 °C for at least 1 h and occasionally more DMSO had to be added to achieve a similar and well applicable consistency. After dissolving, the GPE of each sample was coated on the graphite layered counter electrodes using a doctor blade with a wet layer thickness of 30 μm. Three unsealed DSSCs were assembled per sample by placing the front electrodes on the GPE layer and attaching them with transparent tape. The DSSCs were deliberately not sealed, as the long-term stability of the electrolyte was also to be investigated in terms of possible evaporation.

The measurements of the GPEs with respect to their ionic conductivities were carried out at a temperature of 40 °C with a LWT-01 Voltcraft conductivity pen (Conrad Electronic, Wollerau, Switzerland).

To evaluate the potential benefit of the GPEs regarding their long-term efficiency, measurements of the I-U curves were conducted using a Keithley 2450 source meter (Tektronix Inc., Beaverton, OR, USA) over a maximum of 120 days. They were measured at room temperature under illumination with an AM 1.5 G spectrum and 100 mW/cm$^2$ from a LS0500 solar simulator (LOT-Quantum Design GmbH, Darmstadt, Germany) with a black background to prevent light reflected behind the DSSC from re-entering it. After the measurements, the efficiencies were calculated and the results of all DSSCs per sample are shown for the first three weeks and the best DSSC for each sample is shown over 120 days.

In addition, the I-U curves at day 0, day 28, day 59 and day 110 are also presented for those DSSCs.

## 3. Results

Table 2 shows the results of the ionic conductivity measurements for the different GPEs used to build the DSSCs. The value of the reference electrolyte was taken from Reference [27], where it was measured under identical conditions.

**Table 2.** Ionic conductivities of the investigated GPEs. The sample number refers to Table 1.

| Sample Number | Ionic Conductivity in mS/cm |
|:---:|:---:|
| 1 | $2.76 \pm 0.01$ |
| 2 | $2.74 \pm 0.01$ |
| 3 | $2.32 \pm 0.01$ |
| 4 | $1.91 \pm 0.01$ |
| 5 | $3.56 \pm 0.01$ |
| 6 | $3.30 \pm 0.01$ |
| 7 | $2.51 \pm 0.01$ |
| 8 | $3.29 \pm 0.01$ |
| 9 | $2.96 \pm 0.01$ |
| reference | $0.47 \pm 0.01$ [27] |

For each sample, the three assembled DSSCs were studied for 21 days to examine the most efficient DSSC per sample and their reproducibilities. Therefore, the temporal efficiency progressions over the first 21 days of all investigated DSSCs of this paper are depicted in Figure 1. The denomination of GPEs corresponds to Table 1. Regarding samples 1 and 5 (Figure 1a,c) the differences in the efficiencies were relatively minor compared to the deviations in the other samples. Especially large deviations occurred with respect to sample 3 (Figure 1b), sample 4 (Figure 1c), and sample 8 (Figure 1d). The differences in the efficiency curves of identical DSSCs are shown in Figure 1 and indicate the low reproducibility of the DSSCs built here.

Nevertheless, in order to investigate the long-term stability of the DSSCs depending on the appropriate electrolyte system, the most efficient DSSC per sample, indicated with "a", was selected for closer examination. From the I-U curves, short-circuit current ($I_{SC}$), open-circuit voltage ($U_{OC}$) and the maximum power point ($P_{MPP}$) can be estimated. The central point of discussion is the long-term stability of the individual non-toxic GPEs while the DSSC maintains a high level of efficiency in comparison to the other DSSCs tested.

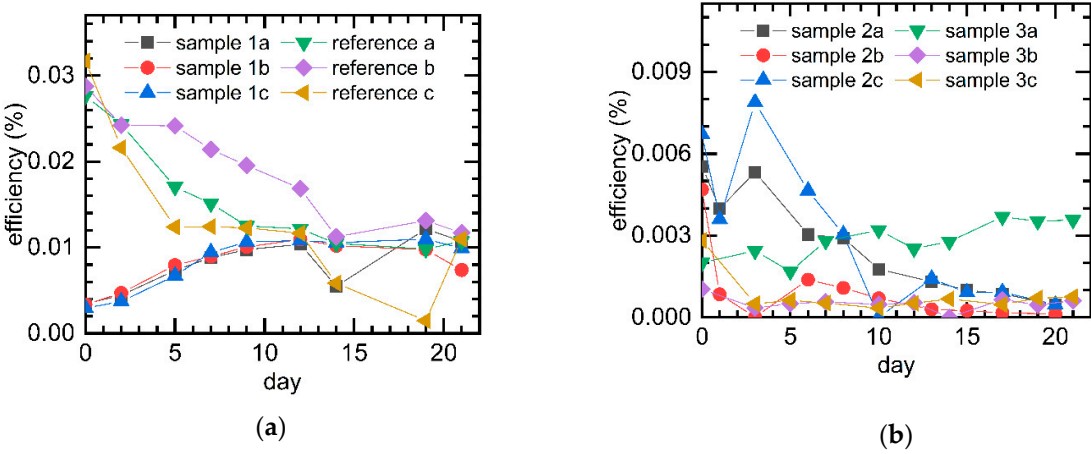

(a)          (b)

**Figure 1.** *Cont.*

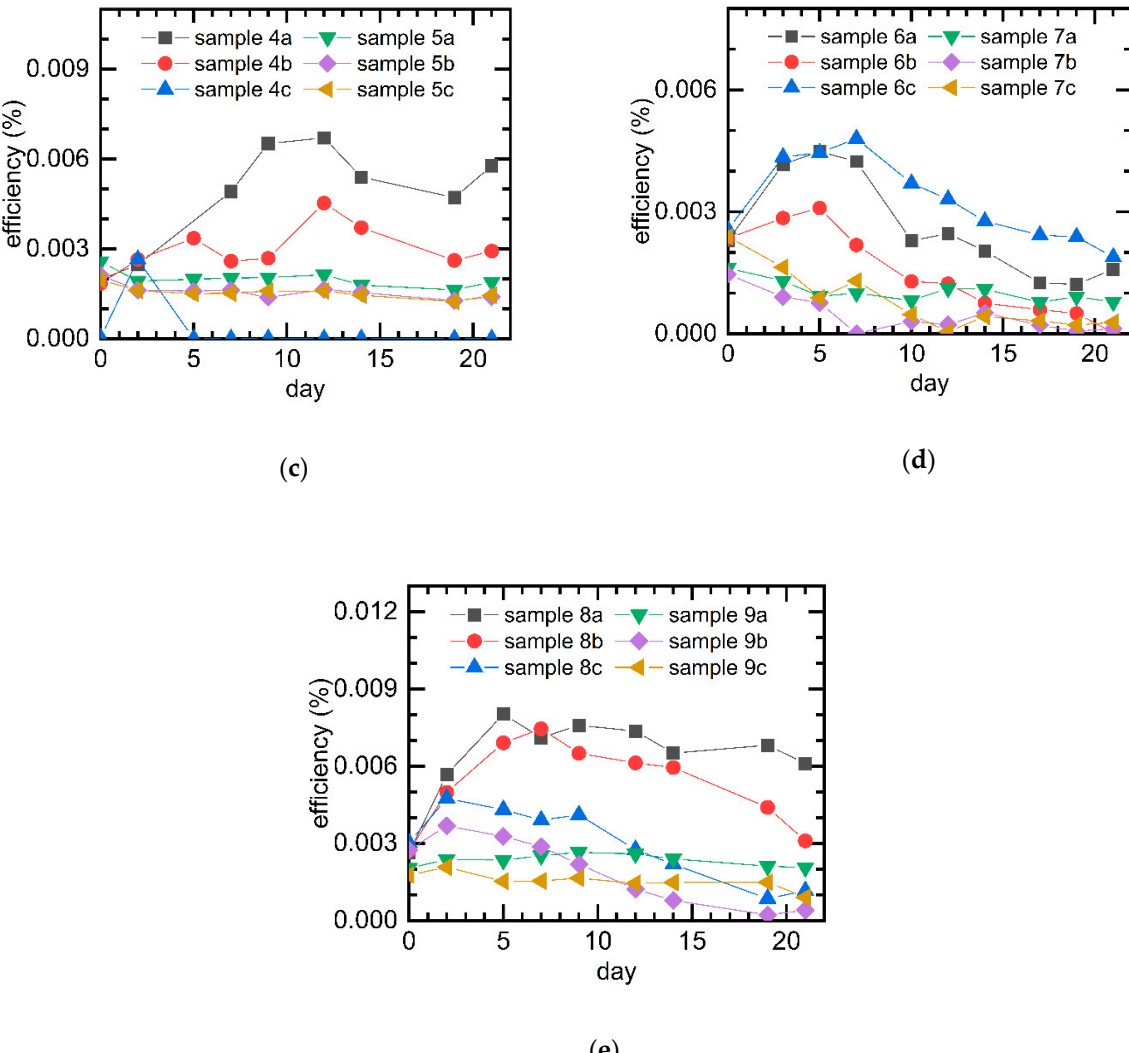

**Figure 1.** Efficiency progressions over the first 21 days of the three assembled DSSCs per sample: (**a**) sample 1 (PAN/PEO GPE) and the reference (commercial liquid electrolyte); (**b**) sample 2 (PAN/PEO GPE with different ratio) and sample 3 (PAN GPE); (**c**) sample 4 (ABS/PEO GPE) and sample 5 (ABS GPE); (**d**) sample 6 (PVA/PEO GPE) and sample 7 (PVA GPE); (**e**) sample 8 (PVDF/PEO GPE) and sample 9 (PVDF GPE). The letters "a" to "c" indicate the different DSSCs per sample, where "a" denotes the best DSSC studied over 120, respectively.

Figure 2 shows the efficiency progression of pure PAN as a gelling agent in the electrolyte, labeled sample 3, in comparison with two blends of PAN/PEO (samples 1 and 2). In this regard, sample 1 contains more PAN, PEO and KI:I$_2$ overall than sample 2, while having a similar ratio. Having the highest DMSO concentration, sample 2 had the highest efficiency during the first few days but decreased rapidly during the first two weeks, with an unsuspected increase after these two weeks and dropping again to values before after three weeks. In contrast, the efficiency of sample 1 increased significantly during this time, peaking highest at around the third week, while sample 3 shows little variation during this period. Noticeable is the efficiency curve of samples 2 and 3 in approximately the second month of measurement, because after a decrease for about two weeks, the efficiency increased again to a similar value as before. Since the efficiency of a DSSC cannot increase apart from the pore filling process [10,26,28,29], in which the efficiency does not decrease before it increases, measurement deviations can be assumed as a possible explanation. In the long run, sample 1 showed the best results in efficiency and long-term stability compared to all other DSSCs containing PAN.

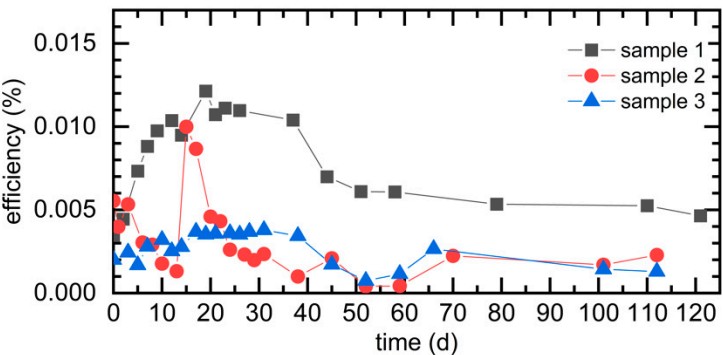

**Figure 2.** Efficiencies of PAN blended with PEO (samples 1 and 2) and PAN (sample 3) GPEs for 120 days.

As an additional opportunity for comparison concerning the efficiency graphs, typical I-U curves are depicted for samples 1 to 3 in Figure 3. The $I_{SC}$ values of samples 1 and 3 increased until day 28 and decreased thereafter, while the $I_{SC}$ of sample 2 decreased until day 28 and then fluctuated slightly. This decreasing trend of sample 2 is also apparent regarding its $U_{OC}$, which corresponds to the efficiency progression in Figure 2. Regarding the $U_{OC}$ of sample 1 an increase until day 28 is again visible with slight fluctuations afterwards and the $U_{OC}$ of sample 3 seems to decrease slightly but constantly. Generally, the higher efficiency of the DSSC of sample 1 (cf. Figure 2) is also evident in the I-U curves depicted in Figure 3.

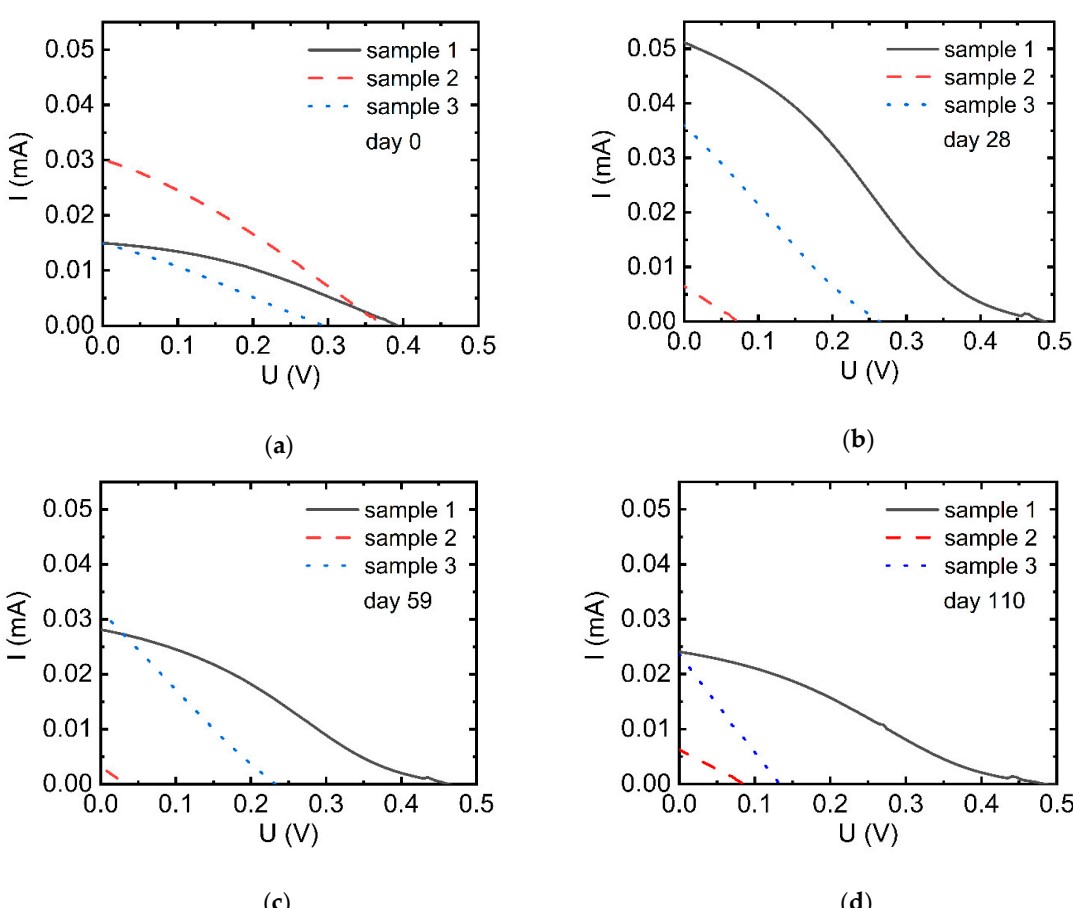

**Figure 3.** I-U curves of PAN blended with PEO (samples 1 and 2) and PAN (sample 3) GPEs for: (**a**) day 0; (**b**) day 28; (**c**) day 59; and (**d**) day 110.

In Figure 4, the efficiencies of the DSSCs with GPEs consisting of ABS, PAN and PVDF, partly blended with PEO, are presented. For all blended GPEs, samples 4, 6 and 8, an increase in efficiency in the first week was identifiable, while the efficiency of the corresponding PEO-less samples (5, 7 and 9) stayed constant or even decreased. DSSCs with a two-component GPE had higher efficiencies than their equivalents without PEO during the entire measurement period. However, except for sample 6, which increased slightly after 100 days, the two-component GPE also dropped over time. In comparison, sample 8, PVDF with PEO, showed the consistently best energy conversion. Sample 4, containing a blend with ABS, had slightly lower efficiencies with a significant decrease in week 8, while PVA blended with PEO (sample 6) fell off after the first week and stayed at low efficiencies.

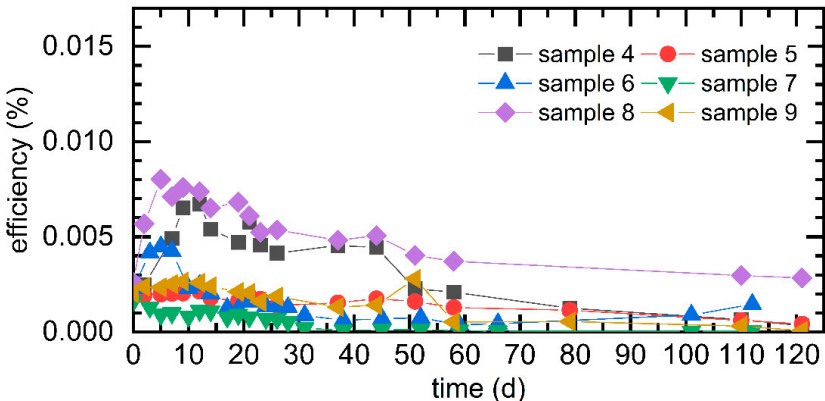

**Figure 4.** Efficiencies of ABS blended with PEO (sample 4), ABS (sample 5), PVA blended with PEO (sample 6), PVA (sample 7) and PVDF blended with PEO (sample 8) and PVDF (sample 9) GPEs for 120 days.

For further comparison, Figure 5 shows the measured I-U curves for days 0, 28, 59 and 110 for the DSSCs of samples 4 to 9. At day 0, the $I_{SC}$ and $U_{OC}$ values were roughly equal (sample 5 shows a higher $I_{SC}$ and sample 7 a lower $U_{OC}$). On day 28 a slight decrease for sample 8 was seen, whereby it always had the highest $P_{MPP}$. Days 28 and 59 showed a further decrease of $U_{OC}$ for sample 4 while $I_{SC}$ rises to a maximum value of 0.065 mA on day 28 and then dropped again. After day 28, the $I_{SC}$ and $U_{OC}$ values of samples 4 to 9 decreased with the aging process, with the values of sample 8 remaining comparatively the highest. In general, the progression of the curves corresponded to the efficiencies depicted in Figure 4 before.

As described above, PEO blended GPEs resulted generally in higher efficiencies and improved long-term stability compared to GPEs without PEO. Therefore, in Figure 6 only the polymer/PEO blends are compared to a reference DSSC with a commercial liquid electrolyte. While all GPE DSSCs started at low and similar efficiency values, the reference DSSC had its highest efficiency at day 0 and was steadily decreasing thereafter. All GPE DSSCs reached a maximum efficiency between the first and third week. Sample 6 (PVA/PEO) and sample 8 (PVDF/PEO) reached values around 0.008% and 0.005% at day 5, sample 4 (ABS/PEO) reached a maximum efficiency of 0.007% at day 13 and the highest value of 0.0125% for sample 1 (PAN/PEO) was reached at day 19. Although all DSSCs decreased after their peak efficiency, their decline was far less than that of the reference. After approximately 50 days, the efficiencies of the GPE DSSCs decreased only slightly until the end of the observed period. It is also worth mentioning that the GPE DSSC efficiency values on day 120 have the same order from highest to lowest like the peak efficiency values reached over the course of the measurement.

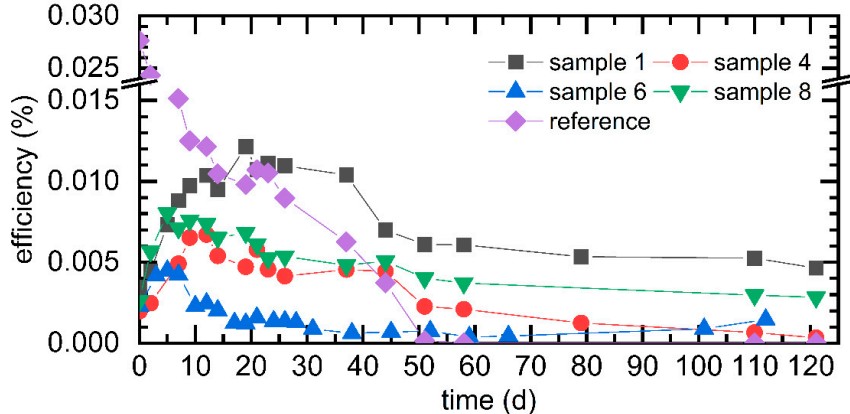

**Figure 5.** I-U curves of ABS blended with PEO (sample 4), ABS (sample 5), PVA blended with PEO (sample 6), PVA (sample 7), and PVDF blended with PEO (sample 8) and PVDF (sample 9) GPEs for: (**a**) day 0; (**b**) day 28; (**c**) day 59; and (**d**) day 110.

**Figure 6.** Efficiencies of PAN blended with PEO (sample 1), ABS blended with PEO (sample 4), PVA blended with PEO (sample 6), PVDF blended with PEO (sample 8) GPEs and a reference DSSC for 120 days.

Figure 7 shows typical I-U curves of the PEO blended GPEs and the reference DSSC. The high efficiency of the liquid electrolyte applied in the reference DSSC on day 0 originated from its high $I_{SC}$ of 0.16 mA, since $U_{OC}$ of all DSSCs had similar values between 0.35 V and 0.42 V on this day. After 28 days the $I_{SC}$ of the reference lowered to 0.064 mA and thus the efficiency was halved. At day 28, all GPE DSSCs reached their peak efficiency and exhibited also a slightly decreasing $P_{MPP}$. At day 59, an overall further decrease of the power conversion ratio was noticed, though the I-U curve of the reference is nearly vanished on this scale. This decreasing trend continued until day 110, whereby sample 1 and 8 were noticeably less affected by it. This is consistent with Figure 6 and confirms that sample 1 with a high proportion of PAN/PEO blend has the best long-term performance, followed by sample 8 (PVDF/PEO). Sample 4 shows a nearly linear progression with the highest value of $I_{SC}$ but only a third of the original $U_{OC}$ measured on day 0. The DSSC containing an ABS/PEO GPE (sample 6) had only a low efficiency after this period.

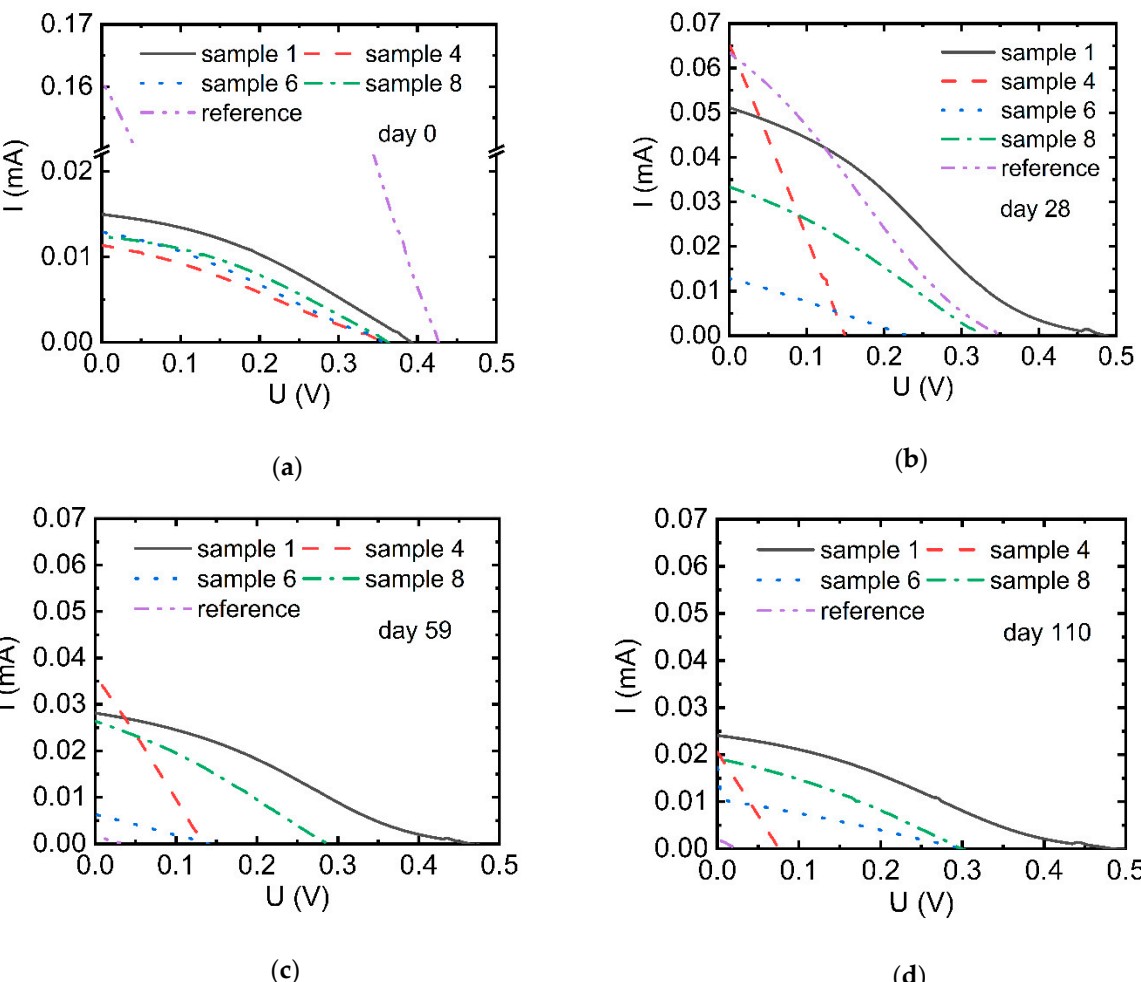

**Figure 7.** I-U curves of PAN blended with PEO (sample 1), ABS blended with PEO (sample 4), PVA blended with PEO (sample 6), PVDF blended with PEO (sample 8) GPEs and a reference for: (**a**) day 0; (**b**) day 28; (**c**) day 59; and (**d**) day 110.

## 4. Discussion

Investigating and presenting the reproducibility of the assembled DSSCs with Figure 1 is necessary to face the known problem of low reproducibility of DSSCs in the greater context of the reproducibility crisis of science [6,28,30–34]. From the observed deviations in the efficiency curves of identical DSSCs per sample over 21 days, it is evident that the DSSCs investigated here have poor reproducibility. Isolated downward deviations in efficiency, e.g., reference DSSC c on day 19 (Figure 1a) or sample 3a on days 5, 12 and 14 (Figure 1b),

are probably due to deviations in measurement, which is probably caused by damaged conductive layers on the electrodes due to the attachment of measuring clamps [27]. The general differences in efficiency indicate deviations in the manufacturing process or in the raw materials of the DSSCs per sample [27]. These can for instance be caused by the pressure, by which the glass electrodes are pressed together, or by the manually applied graphite coatings on the counter electrodes [6,26,27,30]. Even though the best DSSCs were considered for the assessment of long-term stability, the general validity of the results is limited by the low reproducibility. An improvement of the reproducibility is essential for future research.

As is apparent in Figures 2–5, the GPE blends based on PEO show a clearly higher long-term stability and efficiency than the GPEs with only one polymer. In Figures 6 and 7, the measurements of these GPEs mixed with PEO, namely PAN/PEO (sample 1), ABS/PEO (sample 4), PVA/PEO (sample 6) and PVDF/PEO (sample 8), are compared with a reference DSSC. The reference DSSC, which steadily deteriorates (decreasing efficiency) during the measurement, shows a higher energy conversion efficiency than the GPE DSSCs during the first week. The maximum efficiencies of the DSSCs achieved after several days can be explained by the comparatively slower penetration of the nanoporous $TiO_2$/dye layer by the GPE [10,26,28,30]. This pore filling and the resulting contact depends on the structure of the respective polymer [9,35]. Samples 1 and 8 had the least degradation after their peak efficiency, which can also be seen from their relatively constant I-U curves.

According to current literature, the effectiveness of unsealed DSSCs with liquid electrolyte decreases due to leaking and drying of the electrolyte over time [6,7,36]. Thus, because the reference DSSC was not rehydrated during the long-term measurement, a steady reduction in efficiency to a point of malfunction was expected and observed [37]. The positive effect of PEO was already described by Shi et al. [38] before, therefore a good long-term stability of the PEO blended GPEs were also expected and observed. Thanks to the gelling effect of PEO, a stable framework is created, which decreases leakage and evaporation of the GPE [7,11,38,39]. However, GPEs also have some disadvantages, as the contact of the GPEs with the dye and $TiO_2$ layer is worse compared to a liquid electrolyte and increased crystallinity of PEO further reduces the energy conversion efficiency [7,9,11,35,40]. Therefore, it is suggested to blend the polymers PAN, ABS, PVA and PVDF with PEO to adjust the viscosity and reduce the crystallinity of the GPE, which leads to a better penetration of the $TiO_2$ semiconductor, ionic conductivity and overall better energy conversion rate [38,41].

Here, the PAN/PEO blend (sample 1) reaches the highest efficiencies of all GPE DSSCs and shows the best long-term stability. As already mentioned, the types of conductivity of the two polymers are different, but their properties seem to be complementary. The higher ionic conductivity and good gelling properties of PAN [17–19] probably reduce the disadvantages of a pure PEO GPE. An exact comparison of the DSSCs is difficult due to the different additives in the electrolyte and differences in the general structure of the DSSC. Additionally, the problematic reproducibility makes accurate comparisons between DSSCs developed by different researchers difficult [27]. Furthermore, the approach chosen here was to optimize long-term stability and not efficiency.

Another polymer tested for the purpose of GPEs is ABS, shown as pure polymer and blended with PEO. To our knowledge, this is the first time that a combination of these polymers has been tested. As described by Hou et al. [20], ABS is supposed to form a supportive matrix, whereby ABS was mixed with PMMA instead of PEO. There, immiscible ABS coagulates and the butadiene and styrene parts of the polymer form a supportive structure [20,21]. It is to mention that PMMA offers inert ionic transport characteristics in contrast to the coordinating properties of PEO [7,9]. The overall long-term stability and efficiency of the ABS/PEO GPE looks promising, but the decrease in $U_{OC}$ and strong increase of $I_{SC}$ (cf. sample 4 in Figure 5) was unexpected.

A polymer already used for GPEs in combination with PEO and DMSO as solvent is PVA [23,42]. While Teo et al. [42] achieved good long-term stability represented by

the conductivity over 36 days, we see a steady decrease in efficiency after two weeks. It is to mention that, despite similar PEO and PVA weight ratios, the addition of ethylene carbonate, tetrabutylammonium iodide and lithium iodide were also part of the GPE which Teo et al. [42] characterized and therefore the consistency should vary from the GPE used here. Due to our efforts to develop DSSCs that are easy to produce, low-cost and as non-toxic as possible, adding too many additives is not feasible.

Liu et al. [24] described the positive effects of a PVDF/PEO blend and found a mixing ratio of 1:4 (PVDF:PEO) with 3-methoxypropionitrile as solvent and other additives to be optimal, whereas we used a ratio of 1:3 and DMSO as solvent for good coatability. The long-term stability of 500 h (~21 days) shown by them is confirmed and according to our results can also be verified for a period of 120 days (cf. sample 8 in Figure 6). Compared to the mix of PAN and PEO, the value of maximum efficiency is clearly lower, but the course of the curve is straighter overall.

In addition to polymers, a suitable solvent is of great importance, since evaporation leads to a drying of the GPE and will result in an increasing viscosity and decreasing ionic conductivity [43]. The here used non-toxic solvent DMSO with its high vapor pressure offers a stable alternative to the other most often used solvents and is compatible with our goal of producing non-toxic DSSCs [15,41,44].

Comparing the ionic conductivities shown in Table 2 with the composition of the different GPEs given in Table 1, no direct correlation between the electrolyte salt concentration or the polymer content can be seen. Further, comparing the ionic conductivities with the results of the long-term stability measurement, it can be seen that the ionic conductivities do not seem to correlate with the long-term efficiencies. Sorting the ionic conductivities in descending order, the first six samples would be numbers 5, 6, 8, 9, 1 and 2 and while samples 1, 6 and 8 also have high long term stability and efficiency, unfortunately this is not true for numbers 2, 5 and 9. In addition, the purchased reference electrolyte shows a significantly higher efficiency at the beginning compared to the other GPEs, but has a conductivity that is smaller by a factor of four to a factor of seven. One of the reasons why high ionic conductivity does not guarantee good efficiencies is that the contact between the GPE and the nanoporous catalyst and dye layer is very important. Although the molar mass and resulting coil length of the used PEO was constant within the samples, the additional polymers will vary greatly in this regard and, thus, the radius of gyration per GPE, as well as the resulting penetration in the nanoporous catalyst layer will also vary greatly [35,45–49]. Therefore, the contact between the GPE and the dye, which seems to be the limiting factor of the electrolyte in terms of efficiency, will vary depending on the sample. However, a detailed study of this matter is beyond the scope of the paper and will be addressed in future publications.

It worth mentioning that, due to the use of non-toxic, inexpensive materials, a comparison to other laboratory DSSCs large light-active area of 6 cm$^2$ and natural dyes the overall energy conversion efficiencies of the presented DSSCs are low [6,50]. Compared to similar DSSCs from previous publications the here achieved efficiencies are slightly lower [5,26,36,50]. In comparison to earlier publications, more reproducible but also thicker glass plates are now used for the DSSCs, so that higher losses could occur here before the light hits the anthocyanins. Furthermore, the comparatively lower efficiencies are due in particular to observed deviations in the forest fruit tea as the basis for the natural dye, the use of a black background in the measurement and the generally problematic reproducibility [27]. Finally, the comment by Ehrmann and Blachowicz [51] should be mentioned, which notes out that some of the efficiencies of DSSCs with natural dyes found in the literature have unfortunately been calculated too high (mostly by a factor of between 10 and 100) and, thus, place too high an expectation on the efficiencies of non-toxic DSSCs with natural dyes. DSSCs with natural dyes have realistic efficiencies approximately between 0.011% and 0.075%, referring to the highest efficiency of their lifetime without taking aging into account [51].

We are aware of the possibility of degradation of the dye and GPE. The problem of degradation of dye and GPE especially by UV radiation is also addressed in the literature [4,52–55], but their exact statements may vary depending on the test procedure [31]. Besides the exact spectrum of irradiation and environment, as well as binding of the dyes, the PH value and the current temperature influence the degradation [4,31,52–55]. Most attention is paid to commercial dyes such as those based on ruthenium, which remain effective for several thousand hours [31,52]. However, also, the anthocyanins used here, bound to titanium dioxide, are UV-resistant for several hours [53,55], although a direct transfer of the results to the DSSCs shown here is not directly possible. To keep the degradation effects as low as possible, the DSSCs were stored in the dark and only briefly exposed to light for measurement purposes. This of course does not reflect the normal use of a DSSC, but was performed to minimize the aging effects due to UV radiation so that aging due to leakage and evaporation of the electrolyte can be more accurately studied.

## 5. Conclusions

In this paper, further research and development to produce non-toxic DSSCs was presented and discussed. The investigation of different polymeric materials, regarding their long-term stability and efficiency is an important step towards textile DSSCs. Therefore, some materials already used for GPEs as well as polymers never used before in this respect, were investigated. As shown, the polymers alone are less suitable for GPEs than in a mixture with PEO, in which they compensate the existing weaknesses of the base polymer PEO. The best results with a long-term stability over 120 days were achieved by the addition of the polymers PAN and PVDF, whereas PVA showed unsatisfactory results. The polymer ABS tested for the first time in this context also shows promising long-term performance, but inexplicable behaviour of $U_{OC}$ and $I_{SC}$ could be problematic.

By and large, promising polymers have been found to produce GPEs for possible thin film applications and textile-based DSSCs. While the here-used weight ratios oriented toward a good coatability, ongoing research may improve their efficiency and investigate the contact of the GPE with the nanoporous catalyst layer depending on the polymers in more detail. Favourable additives could also have a positive effect on the efficiency, which is here comparatively low due to the natural materials. After optimizing the GPE on glass electrodes, the performance in real textile-based DSSCs will be the centre of investigations.

**Author Contributions:** Conceptualization, M.D. and J.L.S.; methodology, M.D., J.L.S., M.S. and S.A.; validation, M.D., J.L.S. and T.G.; formal analysis, M.D. and J.L.S.; investigation, M.D., J.L.S., M.S. and S.A.; data curation, M.D.; writing–original draft preparation, M.D.; writing–review and editing, M.D., J.L.S. and T.G.; visualization, M.D. and T.G.; supervision, M.D. All authors have read and agreed to the published version of the manuscript.

**Funding:** This work was supported by the State of North Rhine-Westphalia (progres-nrw—nanoDSSC-project) and Deutsche Bundesstiftung Umwelt DBU (German Federal Environmental Foundation).

**Institutional Review Board Statement:** Not applicable.

**Informed Consent Statement:** Not applicable.

**Data Availability Statement:** Not applicable.

**Conflicts of Interest:** The authors declare no conflict of interest. The funders had no role in the design of the study; in the collection, analyses, or interpretation of data; in the writing of the manuscript, or in the decision to publish the results.

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
