# Peer review of "Investigation of the Long-Term Stability of Different Polymers and Their Blends with PEO to Produce Gel Polymer Electrolytes for Non-Toxic Dye-Sensitized Solar Cells"

_applsci, doi:10.3390/app11135834_

Round 1

Reviewer 1 Report

The reviewed paper may be, in general, of enough quality to be considered to publish. The topic of the presented research is very interesting and describes the scientific problem of great importance. However, the quality of the laboratory work and presented experiments seem to be of low level. As the authors mentioned several times, the weakest side of the described work is the reproducibility of the results. It is challenging to conclude from the results of such low reproducibility, but surprisingly, the authors did it, and the discussion seems to be the most substantial part of the manuscript. In my opinion, the only part that is missing is the more detailed discussion on the influence of the anthocyanin dyes degradation or/and reactions with the iodine/iodide present in the electrolyte solutions used.

To conclude, I would like to ask the authors to present a more comprehensive discussion on the stability of the anthocyanin dyes in DSSCs devices utilizing iodine/iodide electrolytes, possibly covered by literature data. The results are of low quality, but the discussion is reasonable, so it should be extended to be complete and finished. Let us say that this is a minor revision.

Reviewer 2 Report

This manuscript studies the long-term stability of DSSCs prepared with various polymer-blends consisting of PEO as gel polymer electrolytes. This work is interesting but there are still some concerns that the authors have to address before it can be published,

  1. First of all, the major concern is the efficiencies of these DSSCs were very very low. The authors might have to compare the efficiencies with some other works to convince the readers that these numbers were meaningful.
  2. The paper only provides electrical and stability data. Some characterization of the polymer gel blends would help to strength the quality of this work.

Round 2

Reviewer 2 Report

The authors have answered my previous concerns. I suggest this paper can be published as is.